# Are ResNets Provably Better than Linear Predictors?

**Ohad Shamir**
Department of Computer Science and Applied Mathematics
Weizmann Institute of Science
Rehovot, Israel
`ohad.shamir@weizmann.ac.il`

## Abstract

A residual network (or ResNet) is a standard deep neural net architecture, with state-of-the-art performance across numerous applications. The main premise of ResNets is that they allow the training of each layer to focus on fitting just the residual of the previous layer's output and the target output. Thus, we should expect that the trained network is no worse than what we can obtain if we remove the residual layers and train a shallower network instead. However, due to the non-convexity of the optimization problem, it is not at all clear that ResNets indeed achieve this behavior, rather than getting stuck at some arbitrarily poor local minimum. In this paper, we rigorously prove that arbitrarily deep, nonlinear residual units indeed exhibit this behavior, in the sense that the optimization landscape contains no local minima with value above what can be obtained with a linear predictor (namely a 1-layer network). Notably, we show this under minimal or no assumptions on the precise network architecture, data distribution, or loss function used. We also provide a quantitative analysis of approximate stationary points for this problem. Finally, we show that with a certain tweak to the architecture, training the network with standard stochastic gradient descent achieves an objective value close or better than any linear predictor.

## 1 Introduction

Residual networks (or ResNets) are a popular class of artificial neural networks, providing state-of-the-art performance across numerous applications [He et al., 2016a,b, Kim et al., 2016, Xie et al., 2017, Xiong et al., 2017]. Unlike vanilla feedforward neural networks, ResNets are characterized by skip connections, in which the output of one layer is directly added to the output of some following layer. Mathematically, whereas feedforward neural networks can be expressed as stacking layers of the form

$$\mathbf{y} = g_\Phi(\mathbf{x}) \,,$$

(where $(\mathbf{x}, \mathbf{y})$ is the input-output pair and $\Phi$ are the tunable parameters of the function $g_\Phi$), ResNets are built from "residual units" of the form $\mathbf{y} = f\left(h(\mathbf{x}) + g_\Phi(\mathbf{x})\right)$, where $f, h$ are fixed functions. In fact, it is common to let $f, h$ be the identity [He et al., 2016b], in which case each unit takes the form

$$\mathbf{y} = \mathbf{x} + g_\Phi(\mathbf{x}) \,. \tag{1}$$

Intuitively, this means that in each layer, the training of $f_\Phi$ can focus on fitting just the "residual" of the target $\mathbf{y}$ given $\mathbf{x}$, rather than $\mathbf{y}$ itself. In particular, adding more depth should not harm performance, since we can effectively eliminate layers by tuning $\Phi$ such that $g_\Phi$ is the zero function. Due to this property, residual networks have proven to be very effective in training extremely deep networks, with hundreds of layers or more.

Despite their widespread empirical success, our rigorous theoretical understanding of training residual networks is very limited. Most recent theoretical works on optimization in deep learning (e.g.

Soltanolkotabi et al. [2017], Yun et al. [2018], Soudry and Hoffer [2017], Brutzkus et al. [2017], Ge et al. [2017], Safran and Shamir [2017], Du and Lee [2018] to name just a few examples) have focused on simpler, feedforward architectures, which do not capture the properties of residual networks. Some recent results do consider residual-like elements (see discussion of related work below), but generally do not apply to standard architectures. In particular, we are not aware of any theoretical justification for the basic premise of ResNets: Namely, that their architecture allows adding layers without harming performance. The problem is that training neural networks involves solving a highly non-convex problem using local search procedures. Thus, even though deeper residual networks can *express* shallower ones, it is not at all clear that the training process will indeed converge to such a network (or a better one). Perhaps, when we attempt to train the residual network using gradient-based methods, we might hit some poor local minimum, with a worse error than what can be obtained with a shallower network? This question is the main motivation to our work.

A secondary motivation are several recent results (e.g. Yun et al. [2018], Safran and Shamir [2017], Du et al. [2017], Liang et al. [2018]),which demonstrate how spurious local minima (with value larger than the global minima) do exist in general when training neural networks, even under fairly strong assumptions. Thus, instead of aiming for a result demonstrating that no such minima exist, which might be too good to be true on realistic networks, we can perhaps consider a more modest goal, showing that no such minima exist above a certain (non-trivial) level set. This level set can correspond, for instance, to the optimal value attainable by shallower networks, without the additional residual layers.

In this paper, we study these questions by considering the competitiveness of a simple residual network (composed of an arbitrarily deep, nonlinear residual unit and a linear output layer) with respect to linear predictors (or equivalently, 1-layer networks). Specifically, we consider the optimization problem associated with training such a residual network, which is in general non-convex and can have a complicated structure. Nevertheless, we prove that the optimization landscape has no local minima *with a value higher* than what can be achieved with a linear predictor on the same data. In other words, if we run a local search procedure and reach a local minimum, we are assured that the solution is no worse than the best obtainable with a linear predictor. Importantly, we show this under fairly minimal assumptions on the residual unit, no assumptions on the data distribution (such as linear separability), and no assumption on the loss function used besides smoothness and convexity in the network's output (which is satisfied for losses used in practice). In addition, we provide a quantitative analysis, which shows how every point which is $\epsilon$-close to being stationary in certain directions (see Sec. 2 for a precise definition) can't be more than poly($\epsilon$) worse than any fixed linear predictor.

The results above are geometric in nature. As we explain later on, they do not necessarily imply that standard gradient-based methods will indeed converge to such desirable solutions (for example, since the iterates might diverge). Nevertheless, we also provide an algorithmic result, showing that if the residual architecture is changed a bit, then a standard stochastic gradient descent (SGD) procedure will result in a predictor similar or better than the best linear predictor. This result relies on a simple, but perhaps unexpected reduction to the setting of online learning, and might be of independent interest.

The supplementary material to this paper contains most proofs (Appendix A) and a discussion of how some of our results can be generalized to vector-valued outputs (Appendix B).

**Related Work**

As far as we know, existing rigorous theoretical results on residual networks all pertain to linear networks, which combine linear residual units of the form

$$\mathbf{y} = \mathbf{x} + W\mathbf{x} = (I + W)\mathbf{x} .$$

Although such networks are not used in practice, they capture important aspects of the non-convexity associated with training residual networks. In particular, Hardt and Ma [2016] showed that linear residual networks with the squared loss have no spurious local minima (namely, every local minimum is also a global one). More recently, Bartlett et al. [2018] proved convergence results for gradient descent on such problems, assuming the inputs are isotropic and the target linear mapping is symmetric and positive definite. Showing similar results for non-linear networks is mentioned in Hardt and Ma [2016] as a major open problem. In our paper, we focus on non-linear residual units, but consider only local minima above some level set.

In terms of the setting, perhaps the work closest to ours is Liang et al. [2018], which considers networks which can be written as $\mathbf{x} \mapsto f_S(\mathbf{x}) + f_D(\mathbf{x})$, where $f_S$ is a one-hidden-layer network, and $f_D$ is an arbitrary, possibly deeper network. Under technical assumptions on the data distribution, activations used, network size, and assuming certain classification losses, the authors prove that the training objective is benign, in the sense that the network corresponding to any local minimum has zero classification error. However, as the authors point out, their architecture is different than standard ResNets (which would require a final tunable layer to combine the outputs of $f_S, f_D$), and their results provably do not hold under such an architecture. Moreover, the technical assumptions are non-trivial, do not apply as-is to standard activations and losses (such as the ReLU activation and the logistic loss), and require specific conditions on the data, such as linear separability or a certain low-rank structure. In contrast, we study a more standard residual unit, and make minimal or no assumptions on the network, data distribution, and loss used. On the flip side, we only prove results for local minima above a certain level set, rather than all such points.

Finally, the idea of studying stationary points in non-convex optimization problems, which are above or below some reference level set, has also been explored in some other works (e.g. Ge and Ma [2017]), but under settings quite different than ours.

## 2   Setting and Preliminaries

We start with a few words about basic notation and terminology. We generally use bold-faced letters to denote vectors (assumed to be in column form), and capital letters to denote matrices or functions. $\|\cdot\|$ refers to the Euclidean norm for vectors and spectral norm for matrices, unless specified otherwise. $\|\cdot\|_{Fr}$ for matrices denotes the Frobenius norm (which always upper bounds the spectral norm). For a matrix $M$, $\text{vec}(M)$ refers to the entries of $M$ written as one long vector (according to some canonical order). Given a function $g$ on Euclidean space, $\nabla g$ denotes its gradient and $\nabla^2 g$ denotes its Hessian. A point $\mathbf{x}$ in the domain of a function $g$ is a local minimum, if $g(\mathbf{x}) \leq g(\mathbf{x}')$ for any $\mathbf{x}'$ in some open neighborhood of $\mathbf{x}$. Finally, we use standard $\mathcal{O}(\cdot)$ and $\Theta(\cdot)$ notation to hide constants, and let $\text{poly}(\mathbf{x}_1, \ldots, \mathbf{x}_r)$ refer to an expression which is polynomial in $\mathbf{x}_1, \ldots, \mathbf{x}_r$.

We consider a residual network architecture, consisting of a residual unit as in Eq. (1), composed with a linear output layer, with scalar output[1]:

$$\mathbf{x} \mapsto \mathbf{w}^\top \left( \mathbf{x} + g_\Phi(\mathbf{x}) \right).$$

We will make no assumptions on the structure of each $g_\Phi$, nor on the overall depth of the network which computes it, except that it's last layer is a tunable linear transformation (namely, that $g_\Phi(\mathbf{x}) = V f_\theta(\mathbf{x})$ for some matrix $V$, not necessarily a square one, and parameters $\theta$). This condition follows the "full pre-activation" structure proposed in He et al. [2016b], which was empirically found to be the best-performing residual unit architecture, and is commonly used in practice (e.g. in TensorFlow). We depart from that structure only in that $V$ is fully tunable rather than a convolution, to facilitate and simplify our theoretical study. Under this assumption, we have that given $\mathbf{x}$, the network outputs

$$\mathbf{x} \mapsto \mathbf{w}^\top \left( \mathbf{x} + V f_\theta(\mathbf{x}) \right),$$

parameterized by a vector $\mathbf{w}$, a matrix $V$, and with some (possibly complicated) function $f_\theta$ parameterized by $\theta$.

**Remark 1** (Biases). *We note that this model can easily incorporate biases, namely predictors of the form $\mathbf{x} \mapsto \mathbf{w}^\top \left( \mathbf{x} + V f_\theta(\mathbf{x}) + \mathbf{a} \right) + a$ for some tunable $a, \mathbf{a}$, by the standard trick of augmenting $\mathbf{x}$ with an additional coordinate whose value is always $1$, and assuming that $f_\theta(\mathbf{x})$ outputs a vector with an additional coordinate of value $1$. Since our results do not depend on the data geometry or specifics of $f_\theta$, they would not be affected by such modifications.*

We assume that our network is trained with respect to some data distribution (e.g. an average over some training set $\{\mathbf{x}_i, y_i\}$), using a loss function $\ell(p, y)$, where $p$ is the network's prediction and $y$ is the target value. Thus, we consider the optimization problem

$$\min_{\mathbf{w}, V, \theta} F(\mathbf{w}, V, \theta) := \mathbb{E}_{\mathbf{x}, y} \left[ \ell(\mathbf{w}^\top (\mathbf{x} + V f_\theta(\mathbf{x})); y) \right], \tag{2}$$

where $\mathbf{w}, V, \theta$ are unconstrained. This objective will be the main focus of our paper. In general, this objective is not convex in $(\mathbf{w}, V, \theta)$, and can easily have spurious local minima and saddle points.

In our results, we will make no explicit assumptions on the distribution of $(\mathbf{x}, y)$, nor on the structure of $f_\theta$. As to the loss, we will assume throughout the paper the following:

**Assumption 1.** *For any $y$, the loss $\ell(p, y)$ is twice differentiable and convex in $p$.*

This assumption is mild, and is satisfied for standard losses such as the logistic loss, squared loss, smoothed hinge loss etc. Note that under this assumption, $F(\mathbf{w}, V, \theta)$ is twice-differentiable with respect to $\mathbf{w}, V$, and in particular the function defined as

$$F_\theta(\mathbf{w}, V) := F(\mathbf{w}, V, \theta)$$

(for any fixed $\theta$) is twice-differentiable. We emphasize that throughout the paper, we will *not* assume that $F$ is necessarily differentiable with respect to $\theta$ (indeed, if $f_\theta$ represents a network with non-differentiable operators such as ReLU or the $\max$ function, we cannot expect that $F$ will be differentiable everywhere). When considering derivatives of $F_\theta$, we think of the input as one long vector in Euclidean space (in order specified by vec()), so $\nabla F_\theta$ is a vector and $\nabla^2 F_\theta$ is a matrix.

As discussed in the introduction, we wish to compare our objective value to that obtained by linear predictors. Specifically, we will use the notation

$$F_{lin}(\mathbf{w}) := F(\mathbf{w}, \mathbf{0}, \theta) = \mathbb{E}_{\mathbf{x}, y}\left[\ell(\mathbf{w}^\top \mathbf{x}; y)\right]$$

to denote the expected loss of a *linear* predictor parameterized by the vector $\mathbf{w}$. By Assumption 1, this function is convex and twice-differentiable.

Finally, we introduce the following class of points, which behave approximately like local minima of $F$ with respect to $(\mathbf{w}, V)$, in terms of its first two derivatives:

**Definition 1** ($\epsilon$-SOPSP). *Let $\mathcal{M}$ be an open subset of the domain of $F(\mathbf{w}, V, \theta)$, on which $\nabla^2 F_\theta(\mathbf{w}, V)$ is $\mu_2$-Lipschitz in $(\mathbf{w}, V)$. Then $(\mathbf{w}, V, \theta) \in \mathcal{M}$ is an $\epsilon$-second-order partial stationary point ($\epsilon$-SOPSP) of $F$ on $\mathcal{M}$, if*

$$\|\nabla F_\theta(\mathbf{w}, V)\| \leq \epsilon \quad and \quad \lambda_{\min}(\nabla^2 F_\theta(\mathbf{w}, V)) \geq -\sqrt{\mu_2 \epsilon} \,.$$

Importantly, note that *any* local minimum $(\mathbf{w}, V, \theta)$ of $F$ must be a 0-SOPSP: This is because $(\mathbf{w}, V)$ is a local minimum of the (differentiable) function $F_\theta$, hence $\|\nabla F_\theta(\mathbf{w}, V)\| = 0$ and $\lambda_{\min}(\nabla^2 F_\theta(\mathbf{w}, V)) \geq 0$. Our definition above directly generalizes the well-known notion of $\epsilon$-second-order stationary points (or $\epsilon$-SOSP) [McCormick, 1977, Nesterov and Polyak, 2006, Jin et al., 2017], which are defined for functions which are twice-differentiable in all of their parameters. In fact, our definition of $\epsilon$-SOPSP is equivalent to requiring that $(\mathbf{w}, V)$ is an $\epsilon$-SOSP of $F_\theta$. We need to use this more general definition, because we are not assuming that $F$ is differentiable in $\theta$. Interestingly, $\epsilon$-SOSP is one of the most general classes of points in non-convex optimization, to which gradient-based methods can be shown to converge in poly$(1/\epsilon)$ iterations.

## 3 Competitiveness with Linear Predictors

Our main results are Thm. 3 and Corollary 1 below, which are proven in two stages: First, we show that at any point such that $\mathbf{w} \neq \mathbf{0}$, $\|\nabla F_\theta(\mathbf{w}, V)\|$ is lower bounded in terms of the suboptimality with respect to the best linear predictor (Thm. 1). We then consider the case $\mathbf{w} = \mathbf{0}$, and show that for such points, if they are suboptimal with respect to the best linear predictor, then *either* $\|\nabla F_\theta(\mathbf{w}, V)\|$ is strictly positive, *or* $\lambda_{\min}(\nabla^2 F_\theta(\mathbf{w}, V))$ is strictly negative (Thm. 2). Thus, building on the definition of $\epsilon$-SOPSP from the previous section, we can show that no point which is suboptimal (compared to a linear predictor) can be a local minimum of $F$.

**Theorem 1.** *At any point $(\mathbf{w}, V, \theta)$ such that $\mathbf{w} \neq \mathbf{0}$, and for any vector $\mathbf{w}^*$ of the same dimension as $\mathbf{w}$,*

$$\|\nabla F_\theta(\mathbf{w}, V)\| \geq \frac{F(\mathbf{w}, V, \theta) - F_{lin}(\mathbf{w}^*)}{\sqrt{2\|\mathbf{w}\|^2 + \|\mathbf{w}^*\|^2 \left(2 + \frac{\|V\|^2}{\|\mathbf{w}\|^2}\right)}} \,.$$

The theorem implies that for any point $(\mathbf{w}, V, \theta)$ for which the objective value $F(\mathbf{w}, V, \theta)$ is larger than that of some linear predictor $F_{lin}(\mathbf{w}^*)$, and unless $\mathbf{w} = \mathbf{0}$, its partial derivative with respect to $(\mathbf{w}, V)$ (namely $\nabla F_\theta(\mathbf{w}, V)$) is non-zero, so it cannot be a stationary point with respect to $\mathbf{w}, V$, nor a local minimum of $F$.

The proof of the theorem appears in the supplementary material, but relies on the following key lemma, which we shall state and roughly sketch its proof here:

**Lemma 1.** *Fix some* $\mathbf{w}, V$ *(where* $\mathbf{w} \neq \mathbf{0}$*) and a vector* $\mathbf{w}^*$ *of the same size as* $\mathbf{w}$*. Define the matrix*

$$G = \left( \mathbf{w} - \mathbf{w}^* \; ; \; \frac{1}{\|\mathbf{w}\|^2} \mathbf{w}(\mathbf{w}^*)^\top V \right) .$$

*Then*

$$\langle vec(G), \nabla F_\theta(\mathbf{w}, V) \rangle \;\geq\; F(\mathbf{w}, V, \theta) - F_{lin}(\mathbf{w}^*) .$$

In other words, the inner product of the gradient with some carefully-chosen vector is lower bounded by the suboptimality of $F(\mathbf{w}, V, \theta)$ compared to a linear predictor (and in particular, if the point is suboptimal, the gradient cannot be zero).

*Proof Sketch of Lemma 1.* We have

$$\langle vec(G), \nabla F_\theta(\mathbf{w}, V) \rangle \;=\; \left\langle \mathbf{w} - \mathbf{w}^* \,,\, \frac{\partial}{\partial \mathbf{w}} F(\mathbf{w}, V, \theta) \right\rangle$$
$$+ \left\langle vec\left( \frac{1}{\|\mathbf{w}\|^2} \mathbf{w}(\mathbf{w}^*)^\top V \right) \,,\, vec\left( \frac{\partial}{\partial V} F(\mathbf{w}, V, \theta) \right) \right\rangle .$$

Let $d_\ell = \frac{\partial}{\partial p} \ell(p; y)|_{p = \mathbf{w}^\top (\mathbf{x} + V f_\theta(\mathbf{x}))}$. A careful technical calculation reveals that the expression above equals

$$\mathbb{E}_{\mathbf{x}, y} \left[ d_\ell \, (\mathbf{w}^*)^\top V f_\theta(\mathbf{x}) \right] + \mathbb{E}_{\mathbf{x}, y} \left[ d_\ell (\mathbf{w} - \mathbf{w}^*)^\top (\mathbf{x} + V f_\theta(\mathbf{x})) \right] .$$

This in turn equals

$$\mathbb{E}_{\mathbf{x}, y} \left[ d_\ell \left( \mathbf{w}^\top (\mathbf{x} + V f_\theta(\mathbf{x})) - (\mathbf{w}^*)^\top \mathbf{x} \right) \right] .$$

Recalling the definition of $d_\ell$, and noting that by convexity of $\ell$, $\frac{\partial}{\partial p} \ell(p; y)(p - \tilde{p}) \geq \ell(p; y) - \ell(\tilde{p}; y)$ for all $p, \tilde{p}$, it follows that the above is lower bounded by

$$\mathbb{E}_{\mathbf{x}, y} \left[ \ell(\mathbf{w}^\top (\mathbf{x} + V f_\theta(\mathbf{x})); y)) - \ell((\mathbf{w}^*)^\top \mathbf{x}; y) \right] \;=\; F(\mathbf{w}, V, \theta) - F_{lin}(\mathbf{w}^*) .$$

$\square$

To analyze the case $\mathbf{w} = \mathbf{0}$, we have the following result:

**Theorem 2.** *For any* $V, \theta, \mathbf{w}^*$,

$$\lambda_{\min} \left( \nabla^2 F_\theta(\mathbf{0}, V) \right) \;\leq\; 0$$

*and*

$$\|\nabla F_\theta(\mathbf{0}, V)\| + \|V\| \sqrt{ |\lambda_{\min} \left( \nabla^2 F_\theta(\mathbf{0}, V) \right)| \cdot \left\| \frac{\partial^2}{\partial \mathbf{w}^2} F_\theta(\mathbf{0}, V) \right\| + \lambda_{\min} \left( \nabla^2 F_\theta(\mathbf{0}, V) \right)^2 }$$
$$\geq \; \frac{F(\mathbf{0}, V, \theta) - F_{lin}(\mathbf{w}^*)}{\|\mathbf{w}^*\|} ,$$

*where* $\lambda_{\min}(M)$ *denotes the minimal eigenvalue of a symmetric matrix* $M$.

Combining the two theorems above, we can show the following main result:

**Theorem 3.** *Fix some positive* $b, r, \mu_0, \mu_1, \mu_2$ *and* $\epsilon \geq 0$, *and suppose* $\mathcal{M}$ *is some convex open subset of the domain of* $F(\mathbf{w}, V, \theta)$ *in which*

- $\max\{\|\mathbf{w}\|, \|V\|\} \leq b$

- $F_\theta(\mathbf{w}, V), \nabla F_\theta(\mathbf{w}, V)$ *and* $\nabla^2 F_\theta(\mathbf{w}, V)$ *are* $\mu_0$-*Lipschitz,* $\mu_1$-*Lipschitz, and* $\mu_2$-*Lipschitz in* $(\mathbf{w}, V)$ *respectively.*

- *For any $(\mathbf{w}, V, \theta) \in \mathcal{W}$, we have $(\mathbf{0}, V, \theta) \in \mathcal{W}$ and $\|\nabla^2 F_\theta(\mathbf{0}, V)\| \leq \mu_1$.*

*Then for any $(\mathbf{w}, V, \theta) \in \mathcal{M}$ which is an $\epsilon$-SOPSP of $F$ on $\mathcal{M}$,*

$$F(\mathbf{w}, V, \theta) \leq \min_{\mathbf{w}: \|\mathbf{w}\| \leq r} F_{lin}(\mathbf{w}) + (\epsilon + \sqrt[4]{\epsilon}) \cdot poly(b, r, \mu_0, \mu_1, \mu_2).$$

We note that the $poly(b, r, \mu_0, \mu_1, \mu_2)$ term hides only dependencies which are at most linear in the individual factors (see the proof in the supplementary material for the exact expression).

As discussed in Sec. 2, any local minima of $F$ must correspond to a 0-SOPSP. Hence, the theorem above implies that for such a point, $F(\mathbf{w}, V, \theta) \leq \min_{\mathbf{w}: \|\mathbf{w}\| \leq r} F_{lin}(\mathbf{w})$ (as long as $F$ satisfies the Lipschitz continuity assumptions for some finite $\mu_0, \mu_1, \mu_2$ on any bounded subset of the domain). Since this holds for any $r$, we have arrived at the following corollary:

**Corollary 1.** *Suppose that on any bounded subset of the domain of $F$, it holds that $F_\theta(\mathbf{w}, V), \nabla F_\theta(\mathbf{w}, V)$ and $\nabla^2 F_\theta(\mathbf{w}, V)$ are all Lipschitz continuous in $(\mathbf{w}, V)$. Then every local minimum $(\mathbf{w}, V, \theta)$ of $F$ satisfies*

$$F(\mathbf{w}, V, \theta) \leq \inf_{\mathbf{w}} F_{lin}(\mathbf{w}).$$

In other words, the objective $F$ has no spurious local minima with value above the smallest attainable with a linear predictor.

**Remark 2** (Generalization to vector-valued outputs). *One can consider a generalization of our setting to networks with vector-valued outputs, namely $\mathbf{x} \mapsto W(\mathbf{x} + V f_\theta(\mathbf{x}))$, where $W$ is a matrix, and with losses $\ell(\mathbf{p}, \mathbf{y})$ taking vector-valued arguments and convex in $\mathbf{p}$ (e.g. the cross-entropy loss). In this more general setting, it is possible to prove a variant of Thm. 1 using a similar proof technique (see Appendix B). However, it is not clear to us how to prove an analog of Thm. 2 and hence Thm. 3. We leave this as a question for future research.*

## 4 Effects of Norm and Regularization

Thm. 3 implies that any $\epsilon$-SOPSP must have a value not much worse than that obtained by a linear predictor. Moreover, as discussed in Sec. 2, such points are closely related to second-order stationary points, and gradient-based methods are known to converge quickly to such points (e.g. Jin et al. [2017]). Thus, it is tempting to claim that such methods will indeed result in a network competitive with linear predictors. Unfortunately, there is a fundamental catch: The bound of Thm. 3 depends on the norm of the point (via $\|\mathbf{w}\|, \|V\|$), and can be arbitrarily bad if the norm is sufficiently large. In other words, Thm. 3 guarantees that a point which is $\epsilon$-SOPSP is only "good" as long as it is not too far away from the origin.

If the dynamics of the gradient method are such that the iterates remain in some bounded domain (or at least have a sufficiently slowly increasing norm), then this would not be an issue. However, we are not a-priori guaranteed that this would be the case: Since the optimization problem is unconstrained, and we are not assuming anything on the structure of $f_\theta$, it could be that the parameters $\mathbf{w}, V$ diverge, and no meaningful algorithmic result can be derived from Thm. 3.

Of course, one option is that this dependence on $\|\mathbf{w}\|, \|V\|$ is an artifact of the analysis, and any $\epsilon$-SOPSP of $F$ is competitive with a linear predictor, regardless of the norms. However, the following example shows that this is not the case:

**Example 1.** *Fix some $\epsilon > 0$. Suppose $\mathbf{x}, \mathbf{w}, V, \mathbf{w}^*$ are all scalars, $\mathbf{w}^* = 1$, $f_\theta(\mathbf{x}) = \epsilon \mathbf{x}$ (with no dependence on a parameter $\theta$), $\ell(p; y) = \frac{1}{2}(p - y)^2$ is the squared loss, and $\mathbf{x} = y = 1$ w.p. 1. Then the objective can be equivalently written as*

$$F(w, v) = \frac{1}{2}(w(1 + \epsilon v) - 1)^2$$

*(see leftmost plot in Figure 1). The gradient and Hessian of $F(w, v)$ equal*

$$\begin{pmatrix} (w - 1 + \epsilon wv)(1 + \epsilon v) \\ (w - 1 + \epsilon wv)\epsilon w \end{pmatrix} \quad and \quad \begin{pmatrix} (1 + \epsilon v)^2 & \epsilon(2w + 2\epsilon wv - 1) \\ \epsilon(2w + 2\epsilon wv - 1) & \epsilon^2 w^2 \end{pmatrix}$$

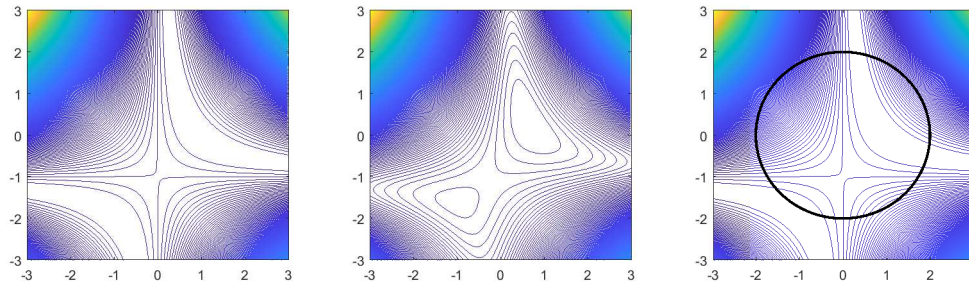

Figure 1: From left to right: Contour plots of (a) $F(w,v) = (w(1+v)-1)^2$, (b) $F(w,v)+\frac{1}{4}(w^2+v^2)$, and (c) $F(w,v)$ superimposed with the constraint $\|(w,v)\| \leq 2$ (inside the circle). The $x$-axis corresponds to $w$, and the $y$-axis corresponds to $v$. Both (b) and (c) exhibit a spurious local minima in the bottom left quadrant of the domain. Best viewed in color.

*respectively. In particular, at $(w,v) = (0,-1/\epsilon)$, the gradient is $\mathbf{0}$ and the Hessian equals $\begin{pmatrix} 0 & -\epsilon \\ -\epsilon & 0 \end{pmatrix}$, which is arbitrarily close to $\mathbf{0}$ if $\epsilon$ is small enough. However, the objective value at that point equals*

$$F\left(0,-\frac{1}{\epsilon}\right) = \frac{1}{2} > 0 = F_{lin}(1).$$

**Remark 3.** *In the example above, $F$ does not have gradients and Hessians with a uniformly bounded Lipschitz constant (over all of Euclidean space). However, for any $\epsilon > 0$, the Lipschitz constants are bounded by a numerical constant over $(w,v) \in [-2/\epsilon, 2/\epsilon]^2$ (which includes the stationary point studied in the construction). This indicates that the problem indeed lies with the norm of $(w,v)$ being unbounded, and not with the Lipschitz constants of the derivatives of $F$.*

One standard approach to ensure that the iterates remain bounded is to add regularization, namely optimize

$$\min_{\mathbf{w},V,\theta} F(\mathbf{w},V,\theta) + R(\mathbf{w},V,\theta) ,$$

where $R$ is a regularization term penalizing large norms of $\mathbf{w}, V, \theta$. Unfortunately, not only does this alter the objective, it might also introduce new spurious local minima that did not exist in $F(\mathbf{w},V,\theta)$. This is graphically illustrated in Figure 1, which plots $F(w,v)$ from Example 1 (when $\epsilon = 1$), with and without regularization of the form $R(w,v) = \frac{\lambda}{2}(w^2+v^2)$ where $\lambda = 1/2$. Whereas the stationary points of $F(w,v)$ are either global minima (along two valleys, corresponding to $\{(w,v) : w(1+\epsilon v) = 1\}$) or a saddle point (at $(w,v) = (1,-1/\epsilon)$), the regularization created a new spurious local minimum around $(w,v) \approx (-1,-1.6)$. Intuitively, this is because the regularization makes the objective value increase well before the valley of global minima of $F$. Other regularization choices can also lead to the same phenomenon. A similar issue can also occur if we impose a hard constraint, namely optimize

$$\min_{\mathbf{w},V,\theta:(\mathbf{w},V,\theta)\in\mathcal{M}} F(\mathbf{w},V,\theta)$$

for some constrained domain $\mathcal{M}$. Again, as Figure 1 illustrates, this optimization problem can have spurious local minima inside its constrained domain, using the same $F$ as before.

Of course, one way to fix this issue is by making the regularization parameter $\lambda$ sufficiently small (or the domain $\mathcal{M}$ sufficiently large), so that the regularization only comes into effect when $\|(w,v)\|$ is sufficiently large. However, the correct choice of $\lambda$ and $\mathcal{M}$ depends on $\epsilon$, and here we run into a problem: If $f_\theta$ is not simply some fixed $\epsilon$ (as in the example above), but changes over time, then we have no a-priori guarantee on how $\lambda$ or $\mathcal{M}$ should be chosen. Thus, it is not clear that any fixed choice of regularization would work, and lead a gradient-based method to a good local minimum.

## 5 Success of SGD Assuming a Skip Connection to the Output

Having discussed the challenges of getting an algorithmic result in the previous section, we now show how such a result is possible, assuming the architecture of our network is changed a bit.

Concretely, instead of the network architecture $\mathbf{x} \mapsto \mathbf{w}^\top(\mathbf{x} + V f_\theta(\mathbf{x}))$, we consider the architecture

$$\mathbf{x} \;\mapsto\; \mathbf{w}^\top \mathbf{x} + \mathbf{v}^\top f_\theta(\mathbf{x}),$$

parameterized by vectors $\mathbf{w}, \mathbf{v}$ and $\theta$, so our new objective can be written as

$$F(\mathbf{w}, \mathbf{v}, \theta) \;=\; \mathbb{E}_{\mathbf{x},y}\left[\ell\left(\mathbf{w}^\top \mathbf{x} + \mathbf{v}^\top f_\theta(\mathbf{x}); y\right)\right].$$

This architecture corresponds to having a skip connection directly to the network's output, rather than to a final linear output layer. It is similar in spirit to the skip-connection studied in Liang et al. [2018], except that they had a two-layer nonlinear network instead of our linear $\mathbf{w}^\top \mathbf{x}$ component.

In what follows, we consider a standard stochastic gradient descent (SGD) algorithm to train our network: Fixing a step size $\eta$ and some convex parameter domain $\mathcal{M}$, we

1. Initialize $(\mathbf{w}_1, \mathbf{v}_1, \theta_1)$ at some point in $\mathcal{M}$
2. For $t = 1, 2, \ldots, T$, we randomly sample a data point $(\mathbf{x}_t, y_t)$ from the underlying data distribution, and perform

$$(\mathbf{w}_{t+1}, \mathbf{v}_{t+1}, \theta_{t+1}) \;=\; \Pi_{\mathcal{M}}\left((\mathbf{w}_t, \mathbf{v}_t, \theta_t) - \eta \nabla h_t(\mathbf{w}_t, \mathbf{v}_t, \theta_t)\right),$$

where

$$h_t(\mathbf{w}, \mathbf{v}, \theta) \;:=\; \ell(\mathbf{w}^\top \mathbf{x}_t + \mathbf{v}^\top f_\theta(\mathbf{x}_t); y_t)$$

and $\Pi_{\mathcal{M}}$ denote an Euclidean projection on the set $\mathcal{M}$.

Note that $h_t(\mathbf{w}, \mathbf{v}, \theta)$ is always differentiable with respect to $\mathbf{w}, \mathbf{v}$, and in the above, we assume for simplicity that it is also differentiable with respect to $\theta$ (if not, one can simply define $\nabla h_t(\mathbf{w}, \mathbf{v}, \theta)$ above to be $\left(\frac{\partial}{\partial \mathbf{w}} h_t(\mathbf{w}, \mathbf{v}, \theta), \frac{\partial}{\partial \mathbf{v}} h_t(\mathbf{w}, \mathbf{v}, \theta), \mathbf{r}_{t,\mathbf{w},\mathbf{v},\theta}\right)$ for some arbitrary vector $\mathbf{r}_{t,\mathbf{w},\mathbf{v},\theta}$, and the result below can still be easily verified to hold).

As before, we use the notation

$$F_{lin}(\mathbf{w}) = \mathbb{E}_{\mathbf{x},y}\left[\ell\left(\mathbf{w}^\top \mathbf{x}; y\right)\right]$$

to denote the expected loss of a linear predictor parameterized by $\mathbf{w}$. The following theorem establishes that under mild conditions, running stochastic gradient descent with sufficiently many iterations results in a network competitive with any fixed linear predictor:

**Theorem 4.** *Suppose the domain $\mathcal{M}$ satisfies the following for some positive constants $b, r, l$:*

- *$\mathcal{M} = \{(\mathbf{w}, \mathbf{v}, \theta) : (\mathbf{w}, \mathbf{v}) \in \mathcal{M}_1, \theta \in \mathcal{M}_2\}$ for some closed convex sets $\mathcal{M}_1, \mathcal{M}_2$ in Euclidean spaces (namely, $\mathcal{M}$ is a Cartesian product of $\mathcal{M}_1, \mathcal{M}_2$).*

- *For any $(\mathbf{x}, y)$ in the support of the data distribution, and any $\theta \in \mathcal{M}_2$, $\ell(\mathbf{w}^\top \mathbf{x} + \mathbf{v}^\top f_\theta(\mathbf{x}); y)$ is $l$-Lipschitz in $(\mathbf{w}, \mathbf{v})$ over $\mathcal{M}_1$, and bounded in absolute value by $r$.*

- *For any $(\mathbf{w}, \mathbf{v}) \in \mathcal{M}_1$, $\sqrt{\|\mathbf{w}\|^2 + \|\mathbf{v}\|^2} \leq b$.*

*Suppose we perform $T$ iterations of stochastic gradient descent as described above, with any step size $\eta = \Theta(b/(l\sqrt{T}))$. Then with probability at least $1 - \delta$, one of the iterates $\{(\mathbf{w}_t, \mathbf{v}_t, \theta_t)\}_{t=1}^T$ satisfies*

$$F(\mathbf{w}_t, \mathbf{v}_t, \theta_t) \;\leq\; \min_{\mathbf{u}:(\mathbf{u},\mathbf{0})\in\mathcal{M}_1} F_{lin}(\mathbf{u}) + \mathcal{O}\left(\frac{bl + r\sqrt{\log(1/\delta)}}{\sqrt{T}}\right).$$

The proof relies on a technically straightforward – but perhaps unexpected – reduction to adversarial online learning, and appears in the supplementary material. Roughly speaking, the idea is that our stochastic gradient descent procedure over $(\mathbf{w}, \mathbf{v}, \theta)$ is equivalent to online gradient descent on $(\mathbf{w}, \mathbf{v})$, with respect to a sequence of functions defined by the iterates $\theta_1, \theta_2, \ldots$. Even though these iterates can change in unexpected and complicated ways, the strong guarantees of online learning (which allow the sequence of functions to be rather arbitrary) allow us to obtain the theorem above.

**Acknowledgements.** We thank the anonymous NIPS 2018 reviewers for their helpful comments. This research is supported in part by European Research Council (ERC) grant 754705.

## Footnotes

[1]See Appendix B for a discussion of how some of our results can be generalized to networks with vector-valued outputs.

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
