[Supplementary Material]

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

# A  Proofs

## A.1  Proof of Lemma 1

To simplify notation, let $d_\ell = \frac{\partial}{\partial p}\ell(p;y)|_{p=\mathbf{w}^\top(\mathbf{x}+Vf_\theta(\mathbf{x}))}$. It is easily verified that

$$\frac{\partial}{\partial \mathbf{w}}F(\mathbf{w},V,\theta) = \mathbb{E}_{\mathbf{x},y}\left[d_\ell(\mathbf{x}+Vf_\theta(\mathbf{x}))\right].$$

Therefore, we have

$$\left\langle \mathbf{w}-\mathbf{w}^*\ ,\ \frac{\partial}{\partial \mathbf{w}}F(\mathbf{w},V,\theta)\right\rangle \ =\ \mathbb{E}_{\mathbf{x},y}\left[d_\ell(\mathbf{w}-\mathbf{w}^*)^\top(\mathbf{x}+Vf_\theta(\mathbf{x}))\right]\ . \tag{3}$$

Proceeding in a similar fashion, it is easily verified that

$$\frac{\partial}{\partial V}F(\mathbf{w},V,\theta) = \mathbb{E}_{\mathbf{x},y}\left[d_\ell\mathbf{w}f_\theta(\mathbf{x})^\top\right],$$

where we write the partial derivative in matrix form. As a result,

$$
\begin{aligned}
&\left\langle \mathrm{vec}\left(\frac{1}{\|\mathbf{w}\|^2}\mathbf{w}(\mathbf{w}^*)^\top V\right)\ ,\ \mathrm{vec}\left(\frac{\partial}{\partial V}F(\mathbf{w},V,\theta)\right)\right\rangle \\
&= \mathrm{trace}\left(\left(\frac{1}{\|\mathbf{w}\|^2}\mathbf{w}(\mathbf{w}^*)^\top V\right)^\top \frac{\partial}{\partial V}F(\mathbf{w},V,\theta)\right) \\
&= \mathbb{E}_{\mathbf{x},y}\left[d_\ell\ \mathrm{trace}\left(\left(\frac{1}{\|\mathbf{w}\|^2}V^\top\mathbf{w}^*\mathbf{w}^\top\right)\mathbf{w}f_\theta(\mathbf{x})^\top\right)\right] \\
&= \mathbb{E}_{\mathbf{x},y}\left[d_\ell\mathrm{trace}\left(V^\top\mathbf{w}^*f_\theta(\mathbf{x})^\top\right)\right] \\
&\overset{(*)}{=} \mathbb{E}_{\mathbf{x},y}\left[d_\ell\ f_\theta(\mathbf{x})^\top V^\top\mathbf{w}^*\right]\ =\ \mathbb{E}_{\mathbf{x},y}\left[d_\ell\ (\mathbf{w}^*)^\top V f_\theta(\mathbf{x})\right]\ , 
\end{aligned}\tag{4}
$$

where in $(*)$ we used the fact that $\mathrm{trace}(ABC) = \mathrm{trace}(CAB)$ for matrices $A,B,C$ that agree in their dimensions.

Using Eq. (3), Eq. (4) and the definition of $G$, it follows that

$$
\begin{aligned}
&\langle \mathrm{vec}(G), \nabla F_\theta(\mathbf{w},V)\rangle \\
&= \left\langle \mathbf{w}-\mathbf{w}^*\ ,\ \frac{\partial}{\partial \mathbf{w}}F(\mathbf{w},V,\theta)\right\rangle \\
&\quad + \left\langle \mathrm{vec}\left(\frac{1}{\|\mathbf{w}\|^2}\mathbf{w}(\mathbf{w}^*)^\top V\right)\ ,\ \mathrm{vec}\left(\frac{\partial}{\partial V}F(\mathbf{w},V,\theta)\right)\right\rangle \\
&= \mathbb{E}_{\mathbf{x},y}\left[d_\ell\ (\mathbf{w}^*)^\top V f_\theta(\mathbf{x})\right] + \mathbb{E}_{\mathbf{x},y}\left[d_\ell(\mathbf{w}-\mathbf{w}^*)^\top(\mathbf{x}+Vf_\theta(\mathbf{x}))\right] \\
&= \mathbb{E}_{\mathbf{x},y}\left[d_\ell(\mathbf{w}^\top(\mathbf{x}+Vf_\theta(\mathbf{x}))-(\mathbf{w}^*)^\top\mathbf{x})\right]\ .
\end{aligned}\tag{5}
$$

Recalling that $d_\ell$ is the derivative of $\ell$ with respect to its first argument when it equals $\mathbf{w}(\mathbf{x}+Vf_\theta(\mathbf{x}))$, and noting that by convexity of $\ell$, $\frac{\partial}{\partial p}\ell(p;y)(p-\tilde{p}) \geq \ell(p;y)-\ell(\tilde{p};y)$ for all $p,\tilde{p}$, it follows that Eq. (5) is lower bounded by

$$\mathbb{E}_{\mathbf{x},y}\left[\ell(\mathbf{w}^\top(\mathbf{x}+Vf_\theta(\mathbf{x}));y))-\ell((\mathbf{w}^*)^\top\mathbf{x};y)\right]\ =\ F(\mathbf{w},V,\theta)-F_{lin}(\mathbf{w}^*)\ .$$

## A.2  Proof of Thm. 1

By Lemma 1 and Cauchy-Schwartz, we have

$$\|G\|_{Fr}\cdot\|\nabla F_\theta(\mathbf{w},V)\|\ \geq\ \|G\|\cdot\|\nabla F_\theta(\mathbf{w},V)\|\ \geq\ F(\mathbf{w},V,\theta)-F_{lin}(\mathbf{w}^*)\ .$$

(where $\|\cdot\|_{Fr}$ denotes the Frobenius norm). Dividing both sides by $\|G\|_{Fr}$, and using the definition of $G$, we get that

$$\|\nabla F_\theta(\mathbf{w},V)\|\ \geq\ \frac{F(\mathbf{w},V,\theta)-F_{lin}(\mathbf{w}^*)}{\sqrt{\|\mathbf{w}-\mathbf{w}^*\|^2+\|\frac{1}{\|\mathbf{w}\|^2}\mathbf{w}(\mathbf{w}^*)^\top V\|_{Fr}^2}}\ . \tag{6}$$

We now simplify this by upper bounding the denominator (note that this leaves the inequality valid regardless of the sign of $F(\mathbf{w}, V, \theta) - F_{lin}(\mathbf{w}^*)$, since if $F(\mathbf{w}, V, \theta) - F_{lin}(\mathbf{w}^*) \geq 0$, this would only decrease its right hand side, and if $F(\mathbf{w}, V, \theta) - F_{lin}(\mathbf{w}^*) < 0$, then the bound remains trivially true since $\|\nabla F_\theta(\mathbf{w}, V)\| \geq 0 > \frac{F(\mathbf{w}, V, \theta) - F_{lin}(\mathbf{w}^*)}{a}$ for any $a > 0$). Specifically, using the facts[2] that $\|AB\| \leq \|A\| \cdot \|B\|$, and that $\|AB\|_{Fr} \leq \|A\|_{Fr} \cdot \|B\|$, we can upper bound the denominator by

$$\sqrt{\|\mathbf{w} - \mathbf{w}^*\|^2 + \frac{1}{\|\mathbf{w}\|^4} \cdot (\|\mathbf{w}\| \cdot \|\mathbf{w}^*\| \cdot \|V\|)^2}.$$

Simplifying the above, using the fact that $\|\mathbf{w} - \mathbf{w}^*\|^2 \leq 2\|\mathbf{w}\|^2 + 2\|\mathbf{w}\|^2$ (as for any vectors $\mathbf{x}, \mathbf{z}$, $\|\mathbf{x} - \mathbf{z}\|^2 \leq \|\mathbf{x}\|^2 + \|\mathbf{z}\|^2 + 2|\mathbf{x}^\top \mathbf{z}| \leq \|\mathbf{x}\|^2 + \|\mathbf{z}\|^2 + (\|\mathbf{x}\|^2 + \|\mathbf{z}\|^2) = 2(\mathbf{x}^2 + \mathbf{z}^2)$), and plugging into Eq. (6), the result follows.

## A.3   Proof of Thm. 2

We will need the following auxiliary lemma:

**Lemma 2.** *Let $M$ be a symmetric real-valued square matrix of the form*

$$M = \begin{pmatrix} b & \mathbf{u}^\top \\ \mathbf{u} & \mathbf{0} \end{pmatrix},$$

*where $b$ is some scalar, $\mathbf{u}$ is a vector, and all entries of $M$ other than the first row and column are $0$. Then the minimal eigenvalue $\lambda_{\min}$ of $M$ is non-positive, and satisfies*

$$\|\mathbf{u}\|^2 = |b\lambda_{\min}| + \lambda_{\min}^2 .$$

*Proof.* By definition, we have $\lambda_{\min} = \min_{\mathbf{z}:\|\mathbf{z}\|=1} \mathbf{z}^\top M \mathbf{z}$. Rewriting $\mathbf{z}$ as $(x; \mathbf{y})$ (where $x$ is the first coordinate of $\mathbf{z}$ and the vector $\mathbf{y}$ represents the other coordinates) and plugging in, this is equivalent to $\min_{x, \mathbf{y}: x^2 + \|\mathbf{y}\|^2 = 1} bx^2 + 2x\mathbf{u}^\top \mathbf{y}$. Clearly, for any fixed $x$, this is minimized when we take $\mathbf{y} = -a\mathbf{u}/\|\mathbf{u}\|$ (for some $a \geq 0$ satisfying the constraints), so we equivalently have $\lambda_{\min} = \min_{x, a: x^2 + a^2 = 1} bx^2 - 2\|\mathbf{u}\|xa$. This is the same as the minimal eigenvalue of the $2 \times 2$ matrix $\begin{pmatrix} b & \|\mathbf{u}\| \\ \|\mathbf{u}\| & 0 \end{pmatrix}$. By standard formulas for $2 \times 2$ matrices, it follows that $\lambda_{\min} = \frac{1}{2}\left(b - \sqrt{b^2 + 4\|\mathbf{u}\|^2}\right)$. Solving for $\|\mathbf{u}\|^2$ and noting that $\lambda_{\min} < 0$, the result follows. $\square$

We now turn to prove the theorem. We will use the shorthand $\ell'$ and $\ell''$ to denote a derivative and second derivative (respectively) of $\ell$ with respect to its first argument. Based on the calculations from earlier, we have

$$\frac{\partial}{\partial \mathbf{w}} F(\mathbf{w}, V, \theta) = \mathbb{E}_{\mathbf{x}, y}\left[\ell'(\mathbf{w}^\top(\mathbf{x} + V f_\theta(\mathbf{x})); y) \cdot (\mathbf{x} + V f_\theta(\mathbf{x}))\right]. \tag{7}$$

$$\frac{\partial}{\partial V} F(\mathbf{w}, V, \theta) = \mathbb{E}_{\mathbf{x}, y}\left[\ell'(\mathbf{w}^\top(\mathbf{x} + V f_\theta(\mathbf{x})); y) \cdot \mathbf{w} f_\theta(\mathbf{x})^\top\right].$$

Therefore, for any indices $i, j$, we have

$$\frac{\partial^2}{\partial V_{i,j} \partial V} F(\mathbf{w}, V, \theta) = \mathbb{E}_{\mathbf{x}, y}\left[\ell''(\mathbf{w}^\top(\mathbf{x} + V f_\theta(\mathbf{x})); y) \cdot (w_i(f_\theta(\mathbf{x}))_j) \cdot \mathbf{w} f_\theta(\mathbf{x})^\top\right]$$

this is $0$ at $\mathbf{w} = \mathbf{0}$, hence

$$\frac{\partial^2}{\partial V^2} F(\mathbf{0}, V, \theta) = \mathbf{0}. \tag{8}$$

Also,

$$\frac{\partial^2}{\partial w_i \partial V} F(\mathbf{w}, V, \theta) = \mathbb{E}_{\mathbf{x}, y}\left[\ell''(\mathbf{w}^\top(\mathbf{x} + V f_\theta(\mathbf{x})); y) \cdot (\mathbf{x} + V f_\theta(\mathbf{x}))_i \cdot \mathbf{w} f_\theta(\mathbf{x})^\top\right]$$
$$+ \mathbb{E}_{\mathbf{x}, y}\left[\ell'(\mathbf{w}^\top(\mathbf{x} + V f_\theta(\mathbf{x})); y) \cdot \mathbf{e}_i f_\theta(\mathbf{x})^\top\right],$$

where $\mathbf{e}_i$ is the $i$-th standard basis vector. At $\mathbf{w} = \mathbf{0}$, this becomes

$$\frac{\partial^2}{\partial w_i \partial V} F(\mathbf{0}, V, \theta) \ = \ \mathbb{E}_{\mathbf{x}, y} \left[ \ell'(0; y) \cdot \mathbf{e}_i f_\theta(\mathbf{x})^\top \right] \ = \ \mathbf{e}_i \mathbf{r}^\top , \tag{9}$$

where we define

$$\mathbf{r} = \mathbb{E}_{\mathbf{x}, y}[\ell'(0; y) f_\theta(\mathbf{x})] . \tag{10}$$

Combining Eq. (8) and Eq. (9), and recalling that $\nabla^2 F_\theta(\mathbf{w}, V)$ is the matrix of second partial derivatives of $F(\mathbf{w}, V, \theta)$ with respect to $(\mathbf{w}, V)$ (viewed as one long vector), we get that

$$\nabla^2 F_\theta(\mathbf{0}, V) \ = \ \begin{pmatrix} & & & \mathbf{r}^\top & \mathbf{0} & \mathbf{0} & \\ & \frac{\partial^2}{\partial \mathbf{w}^2} F_\theta(\mathbf{0}, V) & & \mathbf{0} & \mathbf{r}^\top & \mathbf{0} & \cdots \\ & & & \mathbf{0} & \mathbf{0} & \mathbf{r}^\top & \\ & & & & \vdots & & \\ \mathbf{r} & \mathbf{0} & \mathbf{0} & & & & \\ \mathbf{0} & \mathbf{r} & \mathbf{0} & \cdots & & \mathbf{0} & \\ \mathbf{0} & \mathbf{0} & \mathbf{r} & & & & \\ & \cdots & & & & & \end{pmatrix} .$$

Let $b$ denote the 1st element along the diagonal of $\frac{\partial^2}{\partial \mathbf{w}^2} F(\mathbf{0}, V, \theta)$, and define the matrix

$$M \ = \ \begin{pmatrix} b & \mathbf{r}^\top \\ \mathbf{r} & \mathbf{0} \end{pmatrix} .$$

This is a submatrix of $\nabla^2 F_\theta(\mathbf{0}, V)$, which by Lemma 2, has a minimal eigenvalue $\lambda_{\min}(M) \le 0$, and

$$\|\mathbf{r}\|^2 \ = \ |b\lambda_{\min}(M)| + \lambda_{\min}(M)^2 . \tag{11}$$

Since $M$ is a submatrix of $\nabla^2 F_\theta(\mathbf{0}, V)$, we must also have

$$\lambda_{\min}\left(\nabla^2 F_\theta(\mathbf{0}, V)\right) \le \lambda_{\min}(M) \le 0$$

by the interlacing theorem (proving the first part of the theorem). Finally, since $b$ is an element in the diagonal of $\frac{\partial^2}{\partial \mathbf{w}^2} F(\mathbf{0}, V, \theta)$, we also have $b \le \|\frac{\partial^2}{\partial \mathbf{w}^2} F_\theta(\mathbf{0}, V)\|$. Plugging the above into Eq. (11), we get

$$\|\mathbf{r}\|^2 \ \le \ \left| \lambda_{\min}\left(\nabla^2 F_\theta(\mathbf{0}, V)\right) \right| \cdot \left\| \frac{\partial^2}{\partial \mathbf{w}^2} F_\theta(\mathbf{0}, V) \right\| + \lambda_{\min}\left(\nabla^2 F_\theta(\mathbf{0}, V)\right)^2 . \tag{12}$$

Leaving this equation aside for a moment, we observe that by Eq. (7) and the definition of $\mathbf{r}$ from Eq. (10),

$$\frac{\partial}{\partial \mathbf{w}} F(\mathbf{0}, V, \theta) \ = \ \mathbb{E}_{\mathbf{x}, y} \left[ \ell'(0; y) \cdot (\mathbf{x} + V f_\theta(\mathbf{x})) \right]$$

$$= \ \mathbb{E}_{\mathbf{x}, y} \left[ \ell'(0; y) \mathbf{x} \right] + \mathbb{E}_{\mathbf{x}, y} \left[ \ell'(0; y) V f_\theta(\mathbf{x}) \right] \ = \ \nabla F_{lin}(\mathbf{0}) + V \mathbf{r} ,$$

where we recall that $F_{lin}(\mathbf{w}) = F(\mathbf{w}, \mathbf{0}, \theta) = \mathbb{E}_{\mathbf{x}, y} \left[ \ell(\mathbf{w}^\top \mathbf{x}; y) \right]$. In particular, this implies (by the triangle inequality and Cauchy-Shwartz) that

$$\left\| \frac{\partial}{\partial \mathbf{w}} F(\mathbf{0}, V, \theta) \right\| + \|V\| \cdot \|\mathbf{r}\| \ge \|\nabla F_{lin}(\mathbf{0})\| ,$$

and since $\frac{\partial}{\partial \mathbf{w}} F(\mathbf{0}, V, \theta)$ is a sub-vector of $\nabla F_\theta(\mathbf{0}, V)$, it follows that

$$\|\nabla F_\theta(\mathbf{0}, V)\| + \|V\| \cdot \|\mathbf{r}\| \ \ge \ \|\nabla F_{lin}(\mathbf{0})\| . \tag{13}$$

Fixing some $\mathbf{w}^*$, we have by convexity of $F_{lin}$ that $\nabla F_{lin}(\mathbf{0})^\top (\mathbf{0} - \mathbf{w}^*) \ge F_{lin}(\mathbf{0}) - F_{lin}(\mathbf{w}^*)$, and therefore

$$\|\nabla F_{lin}(\mathbf{0})\| \ \ge \ \frac{F_{lin}(\mathbf{0}) - F_{lin}(\mathbf{w}^*)}{\|\mathbf{w}^*\|} \ = \ \frac{F(\mathbf{0}, V, \theta) - F_{lin}(\mathbf{w}^*)}{\|\mathbf{w}^*\|}$$

for any $V, \theta$. Combining this with Eq. (13) and Eq. (12), and using the shorthand $F_\theta$ for $F_\theta(\mathbf{0}, V)$, we get overall that

$$\|\nabla F_\theta\| + \|V\| \sqrt{\left| \lambda_{\min}\left(\nabla^2 F_\theta(\mathbf{0}, V)\right) \right| \cdot \left\| \frac{\partial^2}{\partial \mathbf{w}^2} F_\theta(\mathbf{0}, V) \right\| + \lambda_{\min}\left(\nabla^2 F_\theta(\mathbf{0}, V)\right)^2} \ \ge \ \frac{F(\mathbf{0}, V, \theta) - F_{lin}(\mathbf{w}^*)}{\|\mathbf{w}^*\|}$$

as required.

## A.4 Proof of Thm. 3

Letting $\mathbf{w}^*$ be a minimizer of $F_{lin}$ over $\{\mathbf{w} : \|\mathbf{w}\| \leq r\}$, and using the shorthand $F_\theta$ for $F_\theta(\mathbf{0}, V)$, we have by Thm. 2 and the assumption in our theorem statement that $\lambda_{\min}(\nabla^2 F_\theta) \leq 0$ that

$$
\begin{aligned}
F(\mathbf{0}, V, \theta) &\leq F_{lin}(\mathbf{w}^*) + r\left(\|\nabla F_\theta\| + b\sqrt{-\mu_1 \lambda_{\min}(\nabla^2 F_\theta) + \lambda_{\min}(\nabla^2 F_\theta)^2}\right) \\
&\leq F_{lin}(\mathbf{w}^*) + r\left(\|\nabla F_\theta\| + b\sqrt{-\lambda_{\min}(\nabla^2 F_\theta)} \cdot \sqrt{\mu_1 - \lambda_{\min}(\nabla^2 F_\theta)}\right) \\
&\leq F_{lin}(\mathbf{w}^*) + r\left(\|\nabla F_\theta\| + b\sqrt{-\lambda_{\min}(\nabla^2 F_\theta)} \cdot \sqrt{\mu_1 + \mu_1}\right) \\
&\leq F_{lin}(\mathbf{w}^*) + r\left(\|\nabla F_\theta\| + \sqrt{2}b\sqrt{-\mu_1 \lambda_{\min}(\nabla^2 F_\theta)}\right) .
\end{aligned}
$$

This inequality refers to $F$ at the point $(\mathbf{0}, V, \theta)$. By the Lipschitz assumptions in our theorem, it implies that for any $\mathbf{w}$ such that $\|\mathbf{w}\| \leq \delta$ for some $\delta > 0$,

$$
F(\mathbf{w}, V, \theta) \leq F_{lin}(\mathbf{w}^*) + \mu_0 \delta + r\left(\|\nabla F_\theta(\mathbf{w}, V)\| + \delta\mu_1 + \sqrt{2}b\sqrt{-\mu_1 \lambda_{\min}(\nabla^2 F_\theta(\mathbf{0}, V))}\right) . \tag{14}
$$

On the other hand, by Thm. 1, for any $\mathbf{w}$ such that $\|\mathbf{w}\| > \delta$, we have

$$
F(\mathbf{w}, V, \theta) \leq F_{lin}(\mathbf{w}^*) + \|\nabla F_\theta(\mathbf{w}, V, \theta)\| \cdot \sqrt{2b^2 + r^2\left(2 + \frac{b^2}{\delta^2}\right)} . \tag{15}
$$

Now, let $(\mathbf{w}, V, \theta)$ be an $\epsilon$-SOPSP (namely, $\|\nabla F_\theta(\mathbf{w}, V)\| \leq \epsilon$ and $\lambda_{\min}(\nabla^2 F_\theta(\mathbf{w}, V)) \geq -\sqrt{\mu_2 \epsilon}$), and note that by the Lipschitz assumptions and Thm. 2,

$$
0 \geq \lambda_{\min}(\nabla^2 F_\theta(\mathbf{0}, V)) \geq \lambda_{\min}(\nabla^2 F_\theta(\mathbf{w}, V) - \mu_2 \|\mathbf{w}\| \geq -\sqrt{\mu_2 \epsilon} - \mu_2 \|\mathbf{w}\| .
$$

Combining this with Eq. (14) and Eq. (15), we get that for any $\delta$, if $(\mathbf{w}, V, \theta)$ is an $\epsilon$-SOPSP, then

$$
\begin{aligned}
F(\mathbf{w}, V, \theta) \leq\ & F_{lin}(\mathbf{w}^*) \\
& + \max\Bigg\{ \mu_0 \delta + r\left(\epsilon + \delta\mu_1 + \sqrt{2}b\sqrt{\mu_1(\sqrt{\mu_2 \epsilon} + \delta\mu_2)}\right) , \\
& \qquad\quad \epsilon\sqrt{2b^2 + r^2\left(2 + \frac{b^2}{\delta^2}\right)} \Bigg\} .
\end{aligned}
$$

In particular, if we pick $\delta = \sqrt{\epsilon}$, we get

$$
\begin{aligned}
F(\mathbf{w}, V, \theta) \leq\ & F_{lin}(\mathbf{w}^*) \\
& + \max\Bigg\{ \mu_0 \sqrt{\epsilon} + r\left(\epsilon + \mu_1 \sqrt{\epsilon} + \sqrt{2}b\sqrt{\mu_1 \sqrt{\epsilon}(\sqrt{\mu_2} + \mu_2)}\right) , \\
& \qquad\quad \sqrt{2b^2\epsilon^2 + r^2\left(2\epsilon^2 + b^2\epsilon\right)} \Bigg\} .
\end{aligned}
$$

Simplifying the above, the right-hand side can be bounded by

$$
F_{lin}(\mathbf{w}^*) + (\epsilon + \sqrt{\epsilon} + \sqrt[4]{\epsilon}) \cdot \text{poly}(b, r, \mu_0, \mu_1, \mu_2) .
$$

Since $\max\{\epsilon, \sqrt[4]{\epsilon}\} \geq \sqrt{\epsilon}$ for any $\epsilon \geq 0$, the result follows.

## A.5 Proof of Thm. 4

Consider any fixed sequence of sampled examples $(\mathbf{x}_1, y_1), (\mathbf{x}_2, y_2), \ldots$. These induce our algorithm to produce a fixed series of iterates $(\mathbf{w}_1, \mathbf{v}_1, \theta_1), (\mathbf{w}_2, \mathbf{v}_2, \theta_2), \ldots$. For any $t$, define the function

$$
g_t((\mathbf{w}, \mathbf{v})) := \ell(\mathbf{w}^\top \mathbf{x}_t + \mathbf{v}^\top f_{\theta_t}(\mathbf{x}_t); y_t) , \tag{16}
$$

where we view $(\mathbf{w}, \mathbf{v})$ as one long vector.

Our first key observation is the following: Our stochastic gradient descent algorithm produces iterates $(\mathbf{w}_t, \mathbf{v}_t)$ which are identical to those produced by the updates

$$
(\mathbf{w}_{t+1}, \mathbf{v}_{t+1}) = \Pi_{\mathcal{M}_1}\left((\mathbf{w}_t, \mathbf{v}_t) - \eta \nabla g_t((\mathbf{w}_t, \mathbf{v}_t))\right) , \tag{17}
$$

where $(\mathbf{w}_1, \mathbf{v}_1) \in \mathcal{M}_1$. This follows from the fact that $\nabla g_t((\mathbf{w}, \mathbf{v})) = \frac{\partial}{\partial(\mathbf{w}, \mathbf{v})} h_t(\mathbf{w}, \mathbf{v}, \theta_t)$ at $(\mathbf{w}, \mathbf{v}) = (\mathbf{w}_t, \mathbf{v}_t)$, and that the projection of $(\mathbf{w}, \mathbf{v}, \theta)$ on the product set $\mathcal{M} = \mathcal{M}_1 \times \mathcal{M}_2$ is equivalent to seperately projecting $(\mathbf{w}, \mathbf{v})$ on $\mathcal{M}_1$ and $\theta$ on $\mathcal{M}_2$.

Our second key observation is that the iterates as defined in Eq. (17) are identical to the iterates produced by an *online* gradient descent algorithm (with step size $\eta$) with respect to the sequence of functions $g_1, g_2, \ldots$ [Zinkevich, 2003, Shalev-Shwartz, 2012, Hazan, 2016]. In particular, the following theorem is well-known (see references above):

**Theorem 5.** *Let $g_1, \ldots, g_T$ be a sequence of convex, differentiable, $l$-Lipschitz functions over a closed convex subset $\mathcal{X}$ of Euclidean space, such that $\mathcal{X} \subseteq \{\mathbf{x} : \|\mathbf{x}\| \leq b\}$. Then if we pick $\mathbf{x}_1 \in \mathcal{X}$ and define $\mathbf{x}_{t+1} = \Pi_{\mathcal{X}}(\mathbf{x}_t - \eta \nabla g_t(\mathbf{x}_t))$ for $t = 1, \ldots, T$, using any $\eta = \Theta(b/(l\sqrt{T}))$, then*

$$\forall \mathbf{x} \in \mathcal{X}, \quad \frac{1}{T} \sum_{t=1}^{T} g_t(\mathbf{x}_t) - \frac{1}{T} \sum_{t=1}^{T} g_t(\mathbf{x}) \leq \mathcal{O}\left(\frac{bl}{\sqrt{T}}\right).$$

In particular, running the gradient descent updates on any sequence of convex Lipschitz functions $g_1, g_2, \ldots$ implies that the average of $g_t(\mathbf{x}_t)$ is not much higher than the minimal value of $\frac{1}{T} \sum_t g_t(\mathbf{x})$ over $\mathbf{x} \in \mathcal{X}$. We now argue that this bound is directly applicable to the functions $g_t$ defined in Eq. (16), over the domain $\mathcal{M}_1$: Indeed, since $\ell$ is convex and assumed to be $l$-Lipschitz in $(\mathbf{w}, \mathbf{v})$, each function $g_t$ is also convex and $l$-Lipschitz. Moreover, the elements in $\mathcal{M}_1$ (viewed as one long vector) are assumed to have an Euclidean norm of at most $b$, and the updates as defined in Eq. (17) are the same as in Thm. 5. Therefore, applying the theorem, we get

$$\forall (\mathbf{w}, \mathbf{v}) \in \mathcal{M}_1, \quad \frac{1}{T} \sum_{t=1}^{T} g_t((\mathbf{w}_t, \mathbf{v}_t)) - \frac{1}{T} \sum_{t=1}^{T} g_t((\mathbf{w}, \mathbf{v})) \leq \mathcal{O}\left(\frac{bl}{\sqrt{T}}\right).$$

We emphasize that this result holds *deterministically* regardless of how the examples $(\mathbf{x}_t, y_t)$ are sampled. Slightly rearranging and plugging back the definition of $g_t$, we get that

$$\forall (\mathbf{w}, \mathbf{v}) \in \mathcal{M}_1, \quad \frac{1}{T} \sum_{t=1}^{T} \left( \ell(\mathbf{w}_t^\top \mathbf{x}_t + \mathbf{v}_t^\top f_{\theta_t}(\mathbf{x}_t); y_t) - \ell(\mathbf{w}^\top \mathbf{x}_t + \mathbf{v}^\top f_{\theta_t}(\mathbf{x}_t); y_t) \right) \leq \mathcal{O}\left(\frac{bl}{\sqrt{T}}\right).$$

In particular, considering any $\mathbf{w}$ such that $(\mathbf{w}, \mathbf{0}) \in \mathcal{M}_1$, we get that

$$\frac{1}{T} \sum_{t=1}^{T} \left( \ell(\mathbf{w}_t \mathbf{x}_t + \mathbf{v}_t^\top f_{\theta_t}(\mathbf{x}_t); y_t) - \ell(\mathbf{w}^\top \mathbf{x}_t; y_t) \right) \leq \mathcal{O}\left(\frac{bl}{\sqrt{T}}\right). \tag{18}$$

Again, this inequality holds deterministically. Now, note that each $(\mathbf{x}_t, y_t)$ is a fresh sample, independent of the history up to round $t$, and conditioned on this history, the expectation of $\ell(\mathbf{w}_t^\top \mathbf{x} + \mathbf{v}_t^\top f_{\theta_t(\mathbf{x}_t)}; y_t) - \ell(\mathbf{w}^\top \mathbf{x}_t; y_t)$ (over $(\mathbf{x}_t, y_t)$) equals $F(\mathbf{w}_t, \mathbf{v}_t, \theta_t) - F_{lin}(\mathbf{w})$. Therefore,

$$\left( F(\mathbf{w}_t, \mathbf{v}_t, \theta_t) - F_{lin}(\mathbf{w}) \right) - \left( \ell(\mathbf{w}_t^\top \mathbf{x} + \mathbf{v}_t^\top f_{\theta_t(\mathbf{x}_t)}; y_t) - \ell(\mathbf{w}^\top \mathbf{x}_t; y_t) \right)$$

is a martingale difference sequence. Moreover, we assume that the losses $\ell$ are bounded by $r$, so by Azuma's inequality, with probability at least $1 - \delta$, the average of the expression above over $t = 1, \ldots, T$ is at most $\mathcal{O}(r\sqrt{\log(1/\delta)/T})$. Combining this with Eq. (18), we get that with probability at least $1 - \delta$,

$$\frac{1}{T} \sum_{t=1}^{T} (F(\mathbf{w}_t, \mathbf{v}_t, \theta_t) - F_{lin}(\mathbf{w})) \leq \mathcal{O}\left(\frac{bl}{\sqrt{T}} + r\sqrt{\frac{\log(1/\delta)}{T}}\right).$$

Rearranging this, we get

$$\frac{1}{T} \sum_{t=1}^{T} F(\mathbf{w}_t, \mathbf{v}_t, \theta_t) \leq F_{lin}(\mathbf{w}) + \mathcal{O}\left(\frac{bl + r\sqrt{\log(1/\delta)}}{\sqrt{T}}\right).$$

Since the left-hand side is an average over $T$ terms, at least one of those terms must be upper bounded by the right-hand side, so there exists some $t$ such that

$$F(\mathbf{w}_t, \mathbf{v}_t, \theta_t) \leq F_{lin}(\mathbf{w}) + \mathcal{O}\left(\frac{bl + r\sqrt{\log(1/\delta)}}{\sqrt{T}}\right).$$

Finally, since this holds for any fixed $\mathbf{w}$ such that $(\mathbf{w}, \mathbf{0}) \in \mathcal{M}_1$, we can choose the one minimizing the right hand side, from which the result follows.

# B  Vector-Valued Networks

In our paper, we focused on the case of neural networks with scalar-valued outputs, and losses over scalar values. However, for tasks such as multi-class classification, it is common to use networks with vector-valued outputs $\mathbf{p}$ (representing a score for each possible class), and losses $\ell(\mathbf{p}, \mathbf{y})$ taking as input vector values (for example, the cross-entropy loss). Thus, it is natural to ask whether our results can be extended to such a setting.

In this setting, our objective function from Eq. (2) takes the form

$$F(W, V, \theta) \ = \ \mathbb{E}_{\mathbf{x},\mathbf{y}}\left[\ell\left(W(\mathbf{x} + V f_\theta(\mathbf{x}))\right); \mathbf{y}\right],$$

where $W$ is a matrix, and as before, we define

$$F_\theta(W, V) \ := \ F(W, V, \theta)$$

where $\theta$ is considered fixed. Using a similar proof technique, it is possible to prove a generalization of Thm. 1 for this case:

**Theorem 6.** *Suppose that $F$ is defined as above, where $\ell$ is differentiable and convex in its first argument. Then at any point $(W, V, \theta)$, for which $W$ has full row rank and minimal singular value $s_{\min}(W) > 0$, it holds that*

$$\|\nabla F_\theta(W, V)\| \ \geq \ \frac{F(W, V, \theta) - F_{lin}(W^*)}{\sqrt{2\|W\|_{Fr}^2 + \|W^*\|_{Fr}^2 \left(2 + \frac{\|V\|^2}{s_{\min}(W)^2}\right)}} \ .$$

*Proof.* The proof proceeds in the same manner as in Thm. 1 (where $W$ was a vector). We will need the following key lemma, whose proof is provided below:

**Lemma 3.** *Fix some $W, V$ and a matrix $W^*$ of the same size as $W$. Define the matrix*

$$G = \left(W - W^* \,;\, W^\top (WW^\top)^{-1} W^* V\right) \ .$$

*Then*

$$\langle vec(G), \nabla F_\theta(W, V) \rangle \ \geq \ F(W, V, \theta) - F_{lin}(W^*) \ .$$

By Lemma 3 and Cauchy-Schwartz, we have

$$\|G\|_{Fr} \cdot \|\nabla F_\theta(W, V)\| \ \geq \ F(W, V, \theta) - F(W^*, \mathbf{0}, \theta).$$

Dividing both sides by $\|G\|_{Fr}$, and using the definition of $G$, we get that

$$\|\nabla F_\theta(W, V)\| \ \geq \ \frac{F(W, V, \theta) - F_{lin}(W^*)}{\sqrt{\|W - W^*\|_{Fr}^2 + \|W^\top(WW^\top)^{-1}W^*V\|_{Fr}^2}} \tag{19}$$

As in the proof of Thm. 1, we can simplify this bound by upper bounding the denominator. Using the facts that $\|AB\| \leq \|A\| \cdot \|B\|$, that $\|AB\|_{Fr} \leq \|A\| \cdot \|B\|_{Fr}$, and that $\|AB\|_{Fr} \leq \|A\|_{Fr}\|B\|$, this denominator is at most

$$\sqrt{\|W - W^*\|_{Fr}^2 + \left(\|W^\top(WW^\top)^{-1}\| \cdot \|W^*\|_{Fr}\|V\|\right)^2}. \tag{20}$$

It is easily verified that if $USV^\top$ is the SVD decomposition of $W$, then $W^\top(WW^\top)^{-1}$ equals $VS(SS)^{-1}U^\top$. Since $V, U$ are orthogonal, it follows that $\|W(WW^\top)^{-1}\| = \|S(SS)^{-1}\| = s_{\min}(W)^{-1}$, where $s_{\min}(W)$ is the smallest singular value of $W$. Moreover, we have $\|W - W^*\|_{Fr}^2 \leq 2\|W\|_{Fr}^2 + 2\|W\|_{Fr}$. Therefore, Eq. (20) is at most

$$\sqrt{2\|W\|_{Fr}^2 + 2\|W^*\|_{Fr}^2 + \left(s_{\min}(W)^{-1}\|W^*\|_{Fr}\|V\|\right)^2}.$$

Slightly simplifying and plugging back into Eq. (19), the result follows. $\qquad\square$

*Proof of Lemma 3.* To simplify notation, let $\mathbf{d}_\ell$ denote the (vector-valued) gradient of $\ell$ with respect to its first argument at the point $W(\mathbf{x} + V f_\theta(\mathbf{x}))$ and $\mathbf{y}$. For any $i, j$, it is easily verified that

$$\frac{\partial}{\partial W_{i,j}} F(W, V, \theta) = \mathbb{E}_{\mathbf{x},\mathbf{y}} \left[ (\mathbf{d}_\ell)_i (\mathbf{x} + V f_\theta(\mathbf{x}))_j \right] ,$$

from which it follows that

$$\frac{\partial}{\partial W} F(W, V, \theta) = \mathbb{E}_{\mathbf{x},\mathbf{y}} \left[ \mathbf{d}_\ell (\mathbf{x} + V f_\theta(\mathbf{x}))^\top \right] ,$$

where we write the partial derivative in matrix form. Therefore, we have

$$\left\langle \text{vec}(W - W^*), \text{vec}\left( \frac{\partial}{\partial W} F(W, V, \theta) \right) \right\rangle = \text{trace}\left( (W - W^*)^\top \frac{\partial}{\partial W} F(W, V, \theta) \right)$$

$$= \mathbb{E}_{\mathbf{x},\mathbf{y}} \left[ \text{trace}\left( (W - W^*)^\top \mathbf{d}_\ell (\mathbf{x} + V f_\theta(\mathbf{x}))^\top \right) \right]$$

$$= \mathbb{E}_{\mathbf{x},\mathbf{y}} \left[ \text{trace}\left( (\mathbf{x} + V f_\theta(\mathbf{x}))^\top (W - W^*)^\top \mathbf{d}_\ell \right) \right]$$

$$= \mathbb{E}_{\mathbf{x},\mathbf{y}} \left[ \mathbf{d}_\ell^\top (W - W^*)(\mathbf{x} + V f_\theta(\mathbf{x})) \right] , \tag{21}$$

where we used the facts that $\text{trace}(ABC) = \text{trace}(CAB)$ and $\text{trace}(A) = \text{trace}(A^\top)$.

Proceeding in a similar fashion, it is straightforward to verify that

$$\frac{\partial}{\partial V_{i,j}} F(W, V, \theta) = \mathbb{E}_{\mathbf{x},\mathbf{y}} \left[ (\mathbf{d}_\ell)^\top W_i (f_\theta(\mathbf{x}))_j \right]$$

(where $W_i$ is the $i$-th column of $W$), and therefore

$$\frac{\partial}{\partial V} F(W, V, \theta) = \mathbb{E}_{\mathbf{x},\mathbf{y}} \left[ W^\top (\mathbf{d}_\ell) f_\theta(\mathbf{x})^\top \right] .$$

As a result,

$$\left\langle \text{vec}(W^\top (WW^\top)^{-1} W^* V , \text{vec}\left( \frac{\partial}{\partial V} F(W, V, \theta) \right) \right\rangle$$

$$= \text{trace}\left( \left( W^\top (WW^\top)^{-1} W^* V \right)^\top \frac{\partial}{\partial V} F(W, V, \theta) \right)$$

$$= \mathbb{E}_{\mathbf{x},\mathbf{y}} \left[ \text{trace}\left( (W^* V)^\top (WW^\top)^{-1} WW^\top (\mathbf{d}_\ell) f_\theta(\mathbf{x})^\top \right) \right]$$

$$= \mathbb{E}_{\mathbf{x},\mathbf{y}} \left[ \text{trace}\left( f_\theta(\mathbf{x})^\top (W^* V)^\top (\mathbf{d}_\ell) \right) \right]$$

$$= \mathbb{E}_{\mathbf{x},\mathbf{y}} \left[ (\mathbf{d}_\ell)^\top W^* V f_\theta(\mathbf{x}) \right] . \tag{22}$$

Summing Eq. (21) and Eq. (22), and recalling the definition of $G$, it follows that

$$\langle \text{vec}(G), \nabla F_\theta(W, V) \rangle =$$

$$= \left\langle \text{vec}(W - W^*), \text{vec}\left( \frac{\partial}{\partial W} F(W, V, \theta) \right) \right\rangle + \left\langle \text{vec}(W^\top (WW^\top)^{-1} W^* V , \text{vec}\left( \frac{\partial}{\partial V} F(W, V, \theta) \right) \right\rangle$$

$$= \mathbb{E}_{\mathbf{x},\mathbf{y}} \left[ \mathbf{d}_\ell^\top (W - W^*)(\mathbf{x} + V f_\theta(\mathbf{x})) \right] + \mathbb{E}_{\mathbf{x},\mathbf{y}} \left[ (\mathbf{d}_\ell)^\top W^* V f_\theta(\mathbf{x}) \right]$$

$$= \mathbb{E}_{\mathbf{x},\mathbf{y}} \left[ (\mathbf{d}_\ell)^\top (W(\mathbf{x} + V f_\theta(\mathbf{x})) - W^* \mathbf{x}) \right] .$$

Recalling that $\mathbf{d}_\ell$ is the gradient of $\ell$ with respect to its first argument when it equals $W(\mathbf{x} + V f_\theta(\mathbf{x}))$, it follows by convexity of $\ell$ that the expression above is lower bounded by

$$\mathbb{E}_{\mathbf{x},\mathbf{y}} \left[ \ell(W(\mathbf{x} + V f_\theta(\mathbf{x}); \mathbf{y})) - \ell(W^* \mathbf{x}; \mathbf{y})) \right] = F(W, V, \theta) - F_{lin}(W^*) .$$

$\square$

This theorem already establishes that our objective in the vector-valued case has no stationary point $(W, V, \theta)$ (with respect to $(W, V)$) with values above $F_{lin}(W^*)$, except possibly when $W$ is not full row rank, or $s_{\min}(W) = 0$. To analyze those cases, one would need an analog of Thm. 2, but unfortunately it is currently unclear how to prove such a result. We leave this question to future research.

Finally, we note that it is straightforward to derive a vector-valued version of Thm. 4 (using stochastic gradient descent over the matrices $W, V$ instead of $\mathbf{w}, \mathbf{v}$), using a virtually identical proof.