[Reviews · NeurIPS 2018]

Reviewer 1



I enjoyed reading the main paper. I tried going over the supplement but couldn't get through the entire document. My score is low because the supplement is not written very well. In particular, lines 455-458 which is a crucial observation, should be explained more or calculations shown even though it is "well known". I do not see any reason why the authors cannot provide a short summary. My second main concern is regarding the usefulness of the result. The generalization bound shown has the standard dependencies on T but in practice the lipschitz constant l is often huge making the bound trivial. I also liked reading section 4. I am not sure why hard constraints are also bad. Can the authors explain lines 261-264 in more detail? Edit: I read the rebuttal from the authors and increased my score. To make the paper more accessible for practitioners, I highly encourage authors to include experiments based on the suggestion provided by R3.

Reviewer 2



=== added after response === I appreciate the authors' honesty in acknowledging some of the weaknesses of the obtained results. I also tend to agree with the authors that the obtained results, relatively speaking, are significant and do shed new insights in understanding ResNet. As such I voted for acceptance (without strong opinion) although the outcome could be either... === end === The main goal of this work is to understand the effect of skip-connections in ResNet, through the lens of optimization. Although ResNet is strictly more powerful than simple linear regression (in the sense that linear regression is a special case of ResNet, if the weights follow a trivial pattern), its optimization may be more challenging than the linear regression special case. The authors formally ruled out this possibility by proving that any local minima of a particular ResNet architecture, or more generally any approximate stationary point, has objective value no larger than that of linear regression. However, finding such a local minima, as the authors showed through a simple example, may still be challenging. Instead, the authors turned to online learning and give a straightforward regret-type guarantee for using SGD to optimize a slightly tweaked ResNet architecture. The paper is very well-written and the technical results, as far as I can tell, are solid. The authors did a terrific job in motivating their problem, which is undoubtedly intricating and significant. The idea to restrict the analysis to stationary points that are sufficiently good (e.g., with objective above a certain value) is not new but nevertheless insightful. However, the analysis also largely overlooked the actual inside structure of the network. Essentially, the authors treated ResNet as a bi-affine function and compared to a linear section. The results, while of some theoretical interest, perhaps are not very relevant in practice. My biggest concern for this work is the following. Take Section 5 for example. First of all, it is misleading to say SGD "converges" for the tweaked ResNet. Secondly, in practice, we can always solve the linear regression problem and then initialize the ResNet weights correspondingly. Then we trivially obtain the regret bound in Theorem 4. (If you do not like the idea of solving linear regression offline first, then run a concurrent online learning algorithm, like SGD, that solves linear regression along with a different algorithm that solves ResNet.) How does this give any insight about the superior empirical performance of ResNet? My point is, Theorem 4, itself really does not tell us much, other than linear regression is a special ResNet. The results in Section 3 are more interesting but again my feeling about it is mixed: on one hand it is certainly encouraging to be able to say something concrete about the loss surface. The conclusion that any local minima is at least as good as the best linear predictor is certainly very nice. But on the other hand, from a practical perspective, to achieve an objective value that is as good as the best linear predictor (for any architecture) is not really difficult. Not to mention that we do not know how to actually get to any of those local minima (in polytime). More technical comments: Theorem 3: are all the Lipschitz constants independent of theta? this is actually where the inside network structure would have an impact and it is perhaps better to put more effort into analyzing it. Corollary 1: why do you need the Lipschitz continuity assumptions here? Does the actual form of the objective function F in Eq (2) matter? If not can we translate the objective so that the linear predictor is a local minima and conclude that all local minima are equally good? (probably not, but why?) Proof of Lemma 2 and Thm 2 can be sharpened: it is easy to see that the minimum eigenvalue of the Hessian after line 416 (appendix) is .5 * [ lambda_{min}(B) - sqrt{ lambda_{min}^2(B) + 4|r|^2 } ], where B is the top-left submatrix (and is positive semidefinite).

Reviewer 3



Summary This paper studies loss surface of residual networks (ResNets). The main contribution of this paper is to prove that any local minimum of risk (empirical or population) of ResNet is better than the risk achieved by the best linear predictor. The authors prove this by upper-bounding the gap between the risk achieved by ResNet and the best linear predictor for “approximate local minima,” which is defined as \epsilon-SOPSP in the paper. For any local minima of ResNet, the upper bound becomes zero, which implies that the local minima is at least as good as (in terms of risk value) the best linear predictor. After that the authors point out that, the bound on the gap depends on the norm of parameters, so the bound is small only if the parameters are close to zero. Thus, unless one can bound the norm of parameters, no meaningful algorithmic result can be derived from the main theorem. They show by an example in which an approximate local minimum has much worse risk than linear predictor, and also that regularization can introduce spurious local minimum. The paper concludes with an algorithmic result that, by tweaking the ResNet architecture, training with stochastic gradient descent achieves a risk value close to (or better than) linear predictors. Comments I like this work a lot. Understanding the loss surface of neural network models is very important, and this theoretical work proves that ANY local minima of ResNet are provably better than linear models. Given that many other existing results require differentiability of local minima, it is very interesting that the main theorem works even for non-differentiable local minima. It is also impressive that they prove this under almost no assumption on data distribution, network architecture, or loss function. The authors are honest about the weakness of the main theorem; they admit that it is hard to get an algorithmic result. However, they partially remedy this issue by tweaking the architecture with almost the same expressive power and show a positive algorithmic theorem. The writing of the paper is very clear and the paper reads smoothly. Overall, it is a well-written paper that contains an interesting result on the energy landscape of ResNets. I recommend acceptance. Minor points - In the proof sketch of Lemma 1, I think it’d be better to be more explicit about the definition of $d\ell$, for example: $d\ell = \frac{\partial}{\partial p} \ell(p;y) \vert_{p = w^T (x+V f_\theta (x))}$. And maybe replace $d\ell$ with some other single character? - There is a missing right parenthesis in the beginning of Theorem 2. ***************Modified on 7/31*************** I’m terribly sorry that it’s already late, but I want to ask the authors (if they have chance to see this) for clarification of whether the ResNet architecture in the paper can be implemented with a stack of standard ResNet units. In the “pre-activation” structure, usually each resnet unit is defined as $t \mapsto t + g_i(t)$, where $g_i(t)$ can be some network, say $g_i (t) = V_i * ReLU(U_i * t)$. If we stack $L > 1$ such residual units, the network would look like h_0 = x, h_i = h_{i-1} + g_i( h_{i-1} ) for i = 1, … , L, h_{L+1} = w^T * h_L, and the output $h_{L+1}$ can be written as h_{L+1} = w^T [ x + g_1(x) + g_2(h_1) + … + g_L(h_{L-1}) ] = w^T [ x + V_1 * ReLU(U_1 * x) + V_2 * ReLU(U_2 * h_1) + … + V_L * ReLU(U_L * h_{L-1}) ]. In contrast, in this paper, the authors analyze a ResNet $x \mapsto w^T [ x + V f_\theta (x) ]$. To me, it seems difficult to formulate the above network output $h_{L+1}$ into this particular form. Does the theorem in the paper hold for this arbitrarily deep stack of residual units, or does it only hold for *only one* residual unit followed by a linear layer? If the former is the case, can you clarify why it holds for $L>1$ units? If the latter is the case, the paper should in some way emphasize this, because readers can get confused between “$f_\theta$ can have arbitrary structure and depth” and “the entire ResNet can be arbitrarily deep.” A possible option could be to refer to the architecture as a “shallow ResNet.” Again, I’m so sorry that I’m too late for additional questions and comments. If you get to see this modification before the deadline, it would be very helpful if you can provide answers. ***************Modified after rebuttal*************** I would like to thank the authors for generously providing feedback to my last-minute questions. And yes, it’d be more helpful for readers to emphasize (although the authors already put significant efforts on it!) the difference between a residual unit vs network, because some readers may not notice the difference of the terms (unit vs network) and then misunderstand the point of the paper; actually, this almost happened to me. Even though the paper only considers ResNets with a single residual unit, my evaluation of the paper stays the same. This is because this paper pushes forward the understanding of loss surface of ResNets, and many existing theoretical works on (conventional) fully connected neural nets are also limited to one-hidden-layer networks.